# Synthesis and Chemopreventive Potential of 5-FU/Genistein Hybrids on Colorectal Cancer Cells

**DOI:** 10.3390/ph15101299

**Published:** 2022-10-21

**Authors:** Gustavo Moreno-Quintero, Wilson Castrillón-Lopez, Angie Herrera-Ramirez, Andrés F. Yepes-Pérez, Jorge Quintero-Saumeth, Wilson Cardona-Galeano

**Affiliations:** 1Chemistry of Colombian Plants Group, Institute of Chemistry, Faculty of Exact and Natural Sciences, University of Antioquia, Calle 70 No. 52–21, A.A 1226, Medellín 050010, Colombia; 2Medical and Experimental Mycology Group, Corporación para Investigaciones Biológicas, Medellín 050034, Colombia

**Keywords:** 5-FU, genistein, hybrid compounds, colorectal cancer, cytotoxicity, antiproliferative activity, cell cycle arrest, apoptosis, docking studies, MD and MM-PBSA studies

## Abstract

A series of 5-FU-Genistein hybrids were synthesized and their structures were elucidated by spectroscopic analysis. The chemopreventive potential of these compounds was evaluated in human colon adenocarcinoma cells (SW480 and SW620) and non-malignant cell lines (HaCaT and CHO-K1). Hybrid **4a** displayed cytotoxicity against SW480 and SW620 cells with IC_50_ values of 62.73 ± 7.26 µM and 50.58 ± 1.33 µM, respectively; compound **4g** induced cytotoxicity in SW620 cells with an IC_50_ value of 36.84 ± 0.71 µM. These compounds were even more selective than genistein alone, the reference drug (5-FU) and the equimolar mixture of genistein plus 5-FU. In addition, hybrids **4a** and **4g** induced time- and concentration-dependent antiproliferative activity and cell cycle arrest at the S-phase and G2/M. It was also observed that hybrid 4a induced apoptosis in SW620 cells probably triggered by the extrinsic pathway in response to the activation of p53, as evidenced by the increase in the levels of caspases 3/8 and the tumor suppressor protein (Tp53). Molecular docking studies suggest that the most active compound **4a** would bind efficiently to proapoptotic human caspases 3/8 and human Tp53, which in turn could provide valuable information on the biochemical mechanism for the in vitro cytotoxic response of this compound in SW620 colon carcinoma cell lines. On the other hand, molecular dynamics (MD) studies provided strong evidence of the conformational stability of the complex between caspase-3 and hybrid **4a** obtained throughout 100 ns all-atom MD simulation. Molecular mechanics Poisson–Boltzmann surface area (MM-PBSA) analyses of the complex with caspase-3 showed that the interaction between the ligand and the target protein is stable. Altogether, the results suggest that the active hybrids, mainly compound **4a**, might act by modulating caspase-3 activity in a colorectal cancer model, making it a privileged scaffold that could be used in future investigations.

## 1. Introduction

Colorectal cancer (CRC) is the second deadliest and widely diagnosed cancer in the world, accounting for 10% of all cancers [1]. 5-fluorouracil (5-FU) is one of the major anticancer agents used for the treatment of several kinds of cancers including CRC. However, this compound not only exhibits a short half-life and low selectivity but also different grades of toxicity including neurological, gastrointestinal disorders and myelosuppression, among others, which leads to dose limitation and treatment discontinuation [2]. For these reasons, this molecule has been extensively modified to obtain conjugates and hybrid molecules, which may improve its therapeutic potential and selectivity, leading to a decrease in side effects [3]. In addition, several natural products have been evaluated since they exhibit a plethora of biological activities in different in vitro and in vivo models. Thereon, the major metabolite of soy (a plant belonging to the Leguminosae family), genistein, has been widely investigated, as it has exhibited a wide range of biological activities [4] including anti-cancer activity [5,6,7,8,9,10].

Considering the potential of the individual molecules, molecular hybridization has emerged as a promising strategy in medicinal chemistry to combine more than two pharmacophores, aiming the search for new therapeutic alternatives to treat colorectal cancer. The pharmacophores of the hybrid molecules exhibit different biological functions, and it is suggested they may display multiple activities, although each unit does not necessarily act on the same biological target [11,12].

On this matter, different investigations have been carried out to evaluate the potential of different compounds based on 5-FU. Figure 1 illustrates the structures of compounds A-F which have been reported for their biological activities. Firstly, the cytotoxicity of 5-FU/Oxaliplatin hybrids **A1–A8**, named Fuplatin, was evaluated against human cancer cells HeLa, MCF-7, CaCo-2, LoVo, HCT-116, A549 and over normal cells MRC-5. In general, these hybrids, were more active than free drugs (Oxaliplatin and 5-FU) or their physical mixtures. The cytotoxicity of the Fuplatin hybrids was selective against the cancer cells tested, as evidenced by lower IC_50_ values on these cells in comparison to normal human lung cells. Hybrid **A5** exhibited the highest activity over human colon carcinoma HCT-116 cells with an IC_50_ value of 0.13 µM, which was 64-fold lower than oxaliplatin and 41-fold lower than the physical mixture [13]. On the other hand, 5-FU/Chalcone hybrid **B** displayed inhibitory activity against four cancer cell lines with IC_50_ values of 4.68, 10.84, 11.87 and 8.17 µM on Hep-G2, RD, LU-1 and FL, respectively. In addition, this compound exhibited lower toxicity than 5-FU against normal VERO cells, as evidenced by a higher IC_50_ value (5-FU = 3.33 µM vs. hybrid **C** = 15.04 µM) [14]. In comparison to 5-FU, compounds **C_1_** and **C_2_**, (5-FU-F16 hybrids), displayed better cytotoxicity in the human gastric cancer cell line (SGC-7901) than in the human fetal gastric epithelial cell line GES-1, as evidenced by higher selectivity indices at the concentration of 10 µM (**C_1_** = 1.16; **C_2_** = 1.08 vs. 5-FU = 0.74) and 50 µM (**C_2_** = 1.20 vs. 5-FU = 0.88). Moreover, **C_1_** exhibited the strongest cytotoxicity on cancer cells among the hybrids evaluated at 50 µM, exhibiting the lowest IC_50_ values over cancer cells (**C_1_** = 0.597). These authors also reported that hybrid **C_1_** showed an ability to target mitochondria which resulted in cell death, cell cycle arrest and the increase in cellular ROS in cancer cells [15].

Other authors (Szeja et al., 2014 [16]) reported that genistein-2,3-Enopyranosyl conjugate **D** was active against human cell lines HCT-116 and DU-145 with IC_50_ values of 2.25 and 4.64 µM, respectively [16]. In addition, in a different investigation, it was reported that conjugate **E** showed high activity over three ER-positive breast cancer cell lines, MCF-7, 21PT and T47D, with IC_50_ values of 0.8, 0.9 and 0.9 μM respectively. This compound induced apoptosis at the IC_50_ concentrations [17]. Finally, hybrid **F** was evaluated on SW480 cells, and the results showed that 48 h post-treatment with 300 µM it displayed significant activity over SW480 cells, causing a significant reduction in cell viability (40.49%), in a time- and concentration-dependent manner [18]. Many other examples of hybrid molecules based on 5-FU conjugated with both natural and synthetic compounds, together with their biological activities on different models of cancer, were summarized by Cardona et al., 2021 [3].

Considering all these facts and the urgent need of searching for new therapeutic alternatives to treat colorectal cancer, we designed and synthesized a series of 5-FU-Genistein hybrids (Figure 2), which were obtained via click reaction between different genistein-alkylazides and propargyl-5-FU, taking advantage of the fact that the 1,2,3-triazole ring plays an important role in medicinal chemistry due to its pharmacological properties; moreover, the incorporation of this moiety decreases the molecule susceptibility to enzymatic degradation, reduction, hydrolysis, and oxidation [19,20]. Furthermore, we evaluated the biological activity of these compounds using different colon cancer cells to determine the chemopreventive potential of these new molecules against colon cancer.

## 2. Results and Discussion

### 2.1. Chemistry

The synthesis of hybrids began with the obtention of the genistein-bromoalkyl 1a-h via Williamson etherification of genistein with 1,ω-dibromoalkanes (ω = 2, 3, 4, 5, 6, 8, 9 and 12) with yields ranging between 61% and 75% [21,22]. Ultrasound-assisted nucleophilic substitution with sodium azide in the compounds **1a**–**h** led to the formation of the genistein-alkylazides **2a–h** in 78%–90% yields [23,24,25]. On the other hand, reaction of 5-FU with propargyl bromide led to propargyl-5-FU (**3**) with 40% yield [26,27,28]. Finally, click reaction between azide alkylgenistein **2a–h** and the alkyne **3** led to the formation of hybrids 4a-h in 60–84% yields [23,29] (Figure 1). The triazole ring was selected as a strategy to incorporate 5-FU, through the click reaction.

The structures of all compounds were established by a combined study of HRMS-ESI (*m/z*), ^1^H-NMR and ^13^C-NMR spectra. HRMS-ESI (*m*/*z*) spectra showed characteristic [M+H]+ peaks corresponding to their molecular weights. The assignments of all the signals to individual H or C-atoms was performed based on typical δ-values and J-constants. The ^1^H-NMR spectra of hybrids dissolved in DMSO showed around 8.12 ppm, a signal corresponding to 5-FU. The chromone (-O-CH=C-) signal was observed around 8.39 ppm. ^13^C-NMR spectra of the hybrids showed around 180.84 ppm a signal corresponding to the carbonyl groups (C=O) of the chromone. The triazolyl ring exhibited signals around 142 and 124 ppm. Finally, the signal corresponding to F-C-C=O of 5-FU was observed around 140.8 and 139.4 ppm.

### 2.2. Biological Activity

#### 2.2.1. Cytotoxicity of 5-FU/Genistein Hybrids on SW480, SW620, HaCaT and CHO-K1 Cell Lines

Biological activity of the synthesized hybrids and the controls (Genistein; 5-FU, the reference drug and their equimolar mixture) was evaluated in human colon adenocarcinoma cells (SW480 and SW620) and non-malignant cell lines (HaCaT and CHO-K1) through sulforhodamine B assay. Cytotoxicity was expressed as the concentration of drug inhibiting cell growth by 50% (IC_50_). The results are summarized in Table 1. According to the results, hybrid **4a** displayed cytotoxicity against both malignant cells evaluated with IC_50_ values of 62.73 ± 7.26 µM and 50.58 ± 1.33 µM for SW480 and SW620 cells, respectively. These values were slightly lower than in non-malignant cells which favored the selectivity indices. In a similar way, compound **4g** induced cytotoxicity in SW620 cells with IC_50_ value of 36.84 ± 0.71 µM. This compound even displayed selective activity as evidenced by higher IC_50_ values in the non-malignant HaCaT and CHO-K1 cells, with greater selectivity indices (>2.71). This selectivity was even better compared to the parental compounds alone and the equimolar mixture, which could be explained by those physicochemical properties of the hybrid that are defined according to its polarity, while in the mixture, the physicochemical properties of the entities must be considered separately. On the other hand, the pharmacokinetic profile of the hybrid is more predictable and the absorption capacity of one part of the structure can be used to facilitate cell permeation of the whole entity [11]. Similar results regarding cytotoxicity of genistein were reported by Zhang and colleagues (2019) [30] in a different cancer model, showing that this isoflavone reduced the cell viability of liver cancer cell line HepG2 in a dose-dependent manner [30]. Our results, together with the reported investigations about the high toxicity of 5-FU, highlight the importance of using this natural source in the development of new molecules with improved chemopreventive potential. In addition, we could not stablish a clear relationship between the cytotoxicity and the length of the alkyl linker because we did not observe a definite tendency when the chain length increased (increased lipophilicity). Finally, even when dibrominated compounds of 7, 10 and 11 carbon atoms would have given better support to the research of these hybrids as anticancer agents, we could not include them due to internal problems.

#### 2.2.2. 5-FU/Genistein Hybrids Induce Antiproliferative Effect on SW480 and SW620 Cells

The antiproliferative activity of the most active hybrids was evaluated against SW480 (**4a**) and SW620 (**4a** and **4g**) colon cancer cells through sulforhodamine B. According to the results, hybrids **4a** and **4g** induced a significant decrease in the percentages of cell viability of these malignant cells (Figure 3). Furthermore, it is important to notice this was a dose- and time-dependent effect, since they induced activity from day 2 at concentrations higher than 12.5 µM, while from day 4 onwards, this inhibitory effect was even significant at the lowest dose of the hybrid (6.25 µM) (Figure 3). In addition, microscopic observations after treatment revealed that these compounds induced changes in the morphology of the cells, altering size and shape, causing a severe damage, suggesting a process of death. The potential antiproliferative effect of genistein together with some derivatives was evaluated by Polkowski and colleagues (2004) [31] against different human cancer cell lines, including human promyelocytic leukemia (HL-60), colorectal adenocarcinoma (Colo-205), breast cancer (MCF-7) and prostate cancer (PC-3), using concentrations between 1 and 100 mM. According to these authors, genistein and its derivatives caused cytostatic and cytotoxic effect in the tested cells, with HL-60 and MCF-7 cells being more susceptible to genistein and its glycosides in comparison to the prostate cancer cells and colorectal adenocarcinoma cell lines. On the other hand, Antosiak and colleagues (2016) [32] evaluated genistein alone or in combination with genistein-8-C-glucoside (G8CG) on the proliferation of cultured human SK-OV-3 ovarian carcinoma cells through the MTT method. They reported that genistein and G8CG significantly decreased cell viability 24 and 48 h post treatment, causing a dose- and time-dependent effect. All these findings highlight the chemopreventive potential of genistein in combination with different other scaffolds such us 5-FU, pointing out the importance of using them in different investigations, contributing to the design of new molecules with great potential and better selectivity against cancer cells, in comparison to conventional chemotherapy.

#### 2.2.3. Effect of Hybrids **4a** and **4g** on Cell Cycle Distribution

With the aim of complementing the results concerning antiproliferative activity, the effect of hybrids **4a** and **4g** on cell cycle distribution of malignant SW480 and SW620 cells was evaluated. Cells were stained with propidium iodide (PI) and they were analyzed by flow cytometry 48 h post-treatment. PI is a fluorescent dye which serves to identify the proportion of cells in one of the three interphase stages, since it intercalates into the major groove of double-stranded DNA. The control vehicle used was DMSO (1%) and the compounds were administered using the IC_50_ value for each cell line (the same concentration was used in all experiments). A representative histogram for each cell line is shown in Figure 4. In SW480 cells (Figure 4A), we observed an important decrease in the population of G0/G1 after treatment with hybrid **4a**, with a consequent increase in S-phase and G2/M (Figure 4B). In addition, we did not observe significant changes in the proportion of cells in the death process (Sub G0/G1) regarding the control. We obtained similar findings for SW620 cells after treatment with hybrids **4a** and **4g** (Figure 5A). In this cell line, there was an important decrease in the population of cells in S-phase with a slight increase in G0/G1; no significant changes were observed in G2/M and Sub G0/G1 (Figure 5B). These results suggest that hybrids **4a** and **4g** could also act as cytostatic agents in these malignant cells. Other researchers have also reported that genistein induces cell cycle arrest of human ovarian cancer cells at the G2/M phase, via activation of DNA damage checkpoint pathways [33]. On the other hand, Zhang et al. (2019) [30] reported that genistein caused G2/M cell cycle arrest of HepG2 cancer cells at 100 μM concentration. These findings support the idea that hybrids based on genistein could be promising, even in those cases with resistance to conventional chemotherapy.

#### 2.2.4. Changes in Mitochondrial Membrane Potential (ΔΨm) and Plasma Membrane Integrity

Considering that changes in mitochondrial membrane potential (ΔΨm) precede caspase activation and subsequent apoptosis [34], we evaluated if hybrids **4a** and **4g** induce mitochondrial depolarization in SW480 and SW620 cells. We used a double staining with a lipophilic dye (Dioc6(3)) and propidium iodide. Dioc6(3) accumulates in the mitochondrial matrix and after reduction in ΔΨm is released in the cytosol. PI is not permeant to live cells and is used to evaluate changes in the integrity of the cell membrane [35,36]. According to the results illustrated in Figure 6, hybrids **4a** and **4g** do not induce significant changes in mitochondrial membrane potential, supporting the hypothesis that we proposed that the probable mechanism of these hybrid molecules could be related with the previous cytostatic effect observed. Contrary to what has been observed in this research, Yoon et al. (2000) [37] demonstrated that 90 µM of genistein after 24 h of treatment induced cytochrome c release by reducing the mitochondrial membrane potential. In addition, it was reported that lower concentrations of genistein (15–60 µM) induced apoptosis in murine T-cell lymphoma cell lines via mitochondrial depolarization [38]. Furthermore, Antosiak et al. (2019) [32] reported that genistein and its natural glucoside G8CG, isolated from flowers of Lupinus luteus L., induced mitochondrial depolarization in cultured human SK-OV-3 ovarian carcinoma cells. These findings highlight the importance of using different models to explore in-depth the greatest potential of the different molecules.

#### 2.2.5. Apoptosis Induction by Hybrids **4a** and **4g**

It is essential to maintain homeostasis in all tissues and for this purpose, the plasma membrane plays a key role as a direct barrier against the extracellular environment to preserve this tissue balance. When the integrity of the cell membrane is broken, the result is the death of the cell, which could be related to different types of programmed cell death such as apoptosis, necroptosis, post-apoptotic, among others. We tested if hybrids **4a** and **4g** induce damage in the plasma membrane and possibly cell death, using a double staining with annexin-FITC and propidium iodide. According to the results illustrated in Figure 5, both hybrids caused a loss in membrane integrity, as evidenced by the increase in the population with positive staining for propidium iodide, suggesting that they induce cell death in SW480 (Figure 7A) and SW620 (Figure 7B) cells. In addition, considering the previous results about the changes in ∆Ψm and plasma membrane integrity, we hypothesized that this death effect is independent of mitochondria, since we did not observe mitochondrial depolarization after treatment with the hybrids. These findings partially agree with the investigations made by other authors who reported the same death effect but with participation of this organelle in the process of death. Susuki et al. (2015) [39] demonstrated that the combination of 5-FU and genistein displayed better antitumor effect against human pancreatic cancer cells when compared with either 5-FU or genistein alone; moreover, these authors reported that genistein itself did not induce apoptosis and autophagy but this isoflavone significantly enhanced 5-FU–induced apoptosis and autophagy. On the contrary, Zhang et al. (2019) [30] reported that genistein induced the production of reactive oxygen species and the subsequent apoptosis mediated by the upregulation of cytosolic cytochrome c, Bax, cleaved caspases 3 and 9 expression and downregulation of Bcl-2 ex-pression in HepG2 cells. All these findings suggest that hybrids **4a** and **4g** could be potential candidates as chemopreventive agents in different cancers or as adjuvants in conventional chemotherapy with 5-FU. Thus, it is important to continue exploring this matter.

#### 2.2.6. Effect of 5-FU/Genistein Hybrids on the Expression Apoptotic Biomarkers

As a complement to the previous findings, we evaluated different apoptotic biomarkers in the possible mechanism associated with hybrids **4a** and **4g** in colon cancer cells. We tested the final executioners in the apoptotic process, caspases 3 and −7. These proteases are involved in the typical cell shrinkage, membrane blebbing, and DNA fragmentation of apoptosis. According to the results, we observed a significant increase in the active form of caspase-3 in SW620 cells only after treatment with hybrid **4a** (Figure 8A), supporting the hypothesis that the possible mechanism of this hybrid is related to apoptosis in this metastatic derivative. In addition, we observed that this hybrid also induced a significant increase in the levels of caspase 8 (Figure 8B); this information, together with the previous results about the lack of participation of the mitochondria, suggest that the possible apoptosis is associated with the extrinsic pathway; however, further investigations are needed to confirm this hypothesis. On the other hand, we observed that hybrid **4a** also induced important changes in the levels of the tumor suppressor protein Tp53 (Figure 8C). This protein regulates different cellular processes such as apoptosis [40,41] and it is mutated in malignant SW620 cells; thus, our results suggest that hybrid **4a** could activate the protein in these cells. This hypothesis about the activation of Tp53 has been previously reported by other authors, who have demonstrated that it retains some of the functions and maintains residual DNA-binding ability [42] and, thus, it is possible to activate it both in vitro and in vivo through different mechanisms [42,43]. These results agree with those reported by Taek Hwang et al. (2005), who showed that genistein in combination with 5-FU abolished the up-regulated state of COX-2 and prostaglandin excretion observed in colon cancer cells (HT29) after treatment with this fluoropyrimidine, probably involving the specific activation of AMPK and the up-regulation of p53, p21, and Bax [44]. All these findings, together with the previous analysis, suggest that the apoptotic process of this hybrid could be mediated by the extrinsic pathway in response to the activation of p53. We also evaluated the same biomarkers in SW480 cells; however, we did not observe significant changes regarding the control (data not shown).

### 2.3. Molecular Modeling Studies

According to the biological assays, hybrid **4a** (n = 2) caused a remarkable apoptotic effect in colorectal cancer cells through a probable multi-target cytotoxic mechanism. The in vitro cytotoxic response produced for **4a** in SW620 colon carcinoma cell lines appear to be strongly associated with the down-modulation of both caspase-3 and caspase-8 proteins, as well as down-regulating the p53 function. In this scenario, we performed computational studies with the aim of exploring a possible binding mechanism of caspase-3/8 and mutant p53 proteins for compound **4a** using the docking program Autodock Vina 1.1.2. To accomplish this goal, compound **4a** was docked inside each catalytic domain of X-ray crystallographic structures of caspase-3 (PDB code: 5i9b), caspase-8 (PDB code: 3kjn) and p53 (PDB code: 1tsr) proteins, and their protein–ligand binding affinities (in kcal/mol), together with their binding modes, were estimated (Table 2).

First, for caspase-3, we conducted self-docking simulations in order to validate our AutoDock Vina docking protocol. To that, we carried out a comparison of the binding modes of the re-dock Ac-DEVD-CMK (in cyan, Figure 9A) and their crystallographic binding mode (in red, Figure 9A) deposited in the PDB archive (PDB code: 5i9b) [48]. The results indicated that our docking procedure was able to reproduce the binding mode of the co-crystallized inhibitor Ac-DEVD-CMK with a good root mean square deviation (RMSD) of 1.075 Å, showing a close homology (Figure 9A). This finding indicates a high level of feasibility in our protein–ligand docking protocol. After the docking protocol was validated, compound **4a** was therefore docked into the caspase-3 catalytic domain. We found that hybrid **4a** (in magenta, Figure 9B) not only binds efficiently to caspase-3 with greater binding affinity (−9.7 kcal/mol) than the current inhibitor Ac-DEVD-CMK (−8.2 kcal/mol), but also fits well inside the catalytic cavity of caspase-3. These facts support our experimental evidence that compound **4a** could prevent cell growth and proliferation in colorectal cancer cells by modulating caspase-3 function. Our modeling work also suggested that similar to co-crystallized inhibitor Ac-DEVD-CMK, **4a** binds to caspase-3 through several non-covalent interactions with those essential amino acid residues vital for caspase-3 function (Figure 9C,D) [48]. Thus, a close-view to the 2D ligand-protein interaction plot after the docking procedure showed that **4a** interacts with the residue Arg207 via two hydrogen bonds at distances of 3.03 and 4.87 Å. Similarly, the 5-FU feature creates two hydrogen bonds with residues Trp214 (4.83 Å) and Phe250 (2.41 Å). In addition, both the fused aromatic ring in chromone and the 5-FU portion were able to bind to the caspase-3 via two π-π T-shaped contacts with Trp206 at distances of 4.71 and 5.46 Å. We also noted that **4a** interacts with caspase-3 through one π−cation contact between the positively charged side chain of Arg207 and the 3-aryl substituent on the chromone moiety. Examination of Figure 9C also shows numerous hydrophobic contacts which would play important roles in stabilizing the **4a**/caspase-3 complex upon binding event. These results suggest that four hydrogen bonding interactions and three π-contacts would have an important role in the effective modulation of caspase-3, consequently preventing SW620 cell growth and proliferation.

Similar to the docking procedure for caspase-3, our docking protocol was validated through the self-docking simulation of the co-crystallized inhibitor MMX-9 inside the catalytic pocket of caspase-8. As shown in Figure 10A, our results indicate that our docking procedure was optimal to reproduce the co-crystallized pose of MMX-9 (in blue, Figure 10) deposited in the PDB archive (PDB code: 3kjn) in comparison with the re-dock MMX-9 (green) with a RMSD less than 2 Å (1.161 Å).

The validated docking procedure was then applied to the compound **4a** (in magenta, Figure 10), which not only shows similar binding affinity (−8.7 kcal/mol) for caspase-8 to that inhibitor MMX-9 (−8.9 kcal/mol), but also is well-accommodated in the active site (Figure 10B), and similarly to the co-crystallographic inhibitor MMX-9, it interacts with those key amino acid residues which contribute significantly to the structure, function, folding, and stability of caspase-8 (Figure 10C,D) [45]. These facts support our experimental evidence that compound **4a** could prevent cell growth and proliferation in SW620 colon carcinoma cell lines via the modulation of caspase-8 function. Thus, a close-view to the 2D ligand-protein interaction plot after the docking procedure showed that **4a** interacts with caspase-8 via multiple sites (Figure 10C). First, five intermolecular hydrogen bonds interactions take place during the **4a**-caspase-8 binding event as follows: the carbonyl group moiety and the fluorine halogen atom directly attached to the 5-FU ring interact simultaneously with key residue Gln358 via two intermolecular hydrogen bonds at a distance of 5.16 and 2.93 Å, respectively; the critical residue Cys360 residue formed one hydrogen bond with the fluorine halogen atom (2.72 Å), and both the 5-OH and 7-OR substituents on the chromone moiety formed with Asp455 and Tyr365 two hydrogen bonds at a distance of 4.63 and 5.89 Å, respectively. In addition, one π−cation contact was observed between the positively charged side chain of Arg413 and the 5-FU ring, and two intramolecular π-π T-shaped contacts were observed between the fused aromatic ring in chromone and the important residue Tyr412. It was also noticed that the fluorine atom attached to the 5-FU ring interacted with the Ser316 and Ala359 residues located in the caspase-8 active site. Finally, multiple hydrophobic interactions were observed in the **4a**-caspase-8 complex, which would play important roles in stabilizing the binding complex. The observed interactions between promising **4a** and those key amino acid residues in caspase-8 could provide a reasonable explanation for the experimental observations.

To explore possible binding mechanisms to p53, hybrid **4a** was fitted inside the human p53 core DNA-binding domain. To accomplish this goal, protein–ligand docking simulations of this compound were conducted in the crystal structure of the human p53 in complex with DNA (PDB ID: 1tsr). Likewise, two potent quinazoline-based p53-DNA-binding inhibitors, SCH529074 and NSC194598 NSC194598 were also evaluated. Although the p53/DNA-binding interface is usually composed of several amino acid residues, only ten “hot spot” sites located in the DNA-binding domain of p53 (Lys132, Lys164, Ser240, Ser241, Arg248, Pro250, Val272, Arg273, Thr284 and Glu285) play a critical structural and functional role for the biological function of human p53 [46]. In this scenario, docking experiments were performed positioning **4a** between the H2 helix and L3 loop of p53 protein harboring as many hot-spot residues as possible. The results showed that compound **4a** (in magenta, Figure 11) is not only capable of binding to p53 with significantly better binding affinity (−7.9 kcal/mol) than those current inhibitors SCH529074 (−7.0 kcal/mol; in green) and NSC194598 (−7.3 kcal/mol, in red), but is also well-accommodated into the p53/DNA-binding interface. These facts, along with the experimental evidence, allow us to infer that compound **4a** affects p53 protein function by disrupting p53 DNA binding, thereby preventing cell growth and proliferation in SW620 colon cancer cells. This last statement was also supported by a rigorous visual examination of the 2D protein-ligand interaction diagram upon the docking event, which indicated that similar to those inhibitors SCH529074 and NSC194598, **4a** interact with those critical hot-spot residues involved in the p53/DNA-binding interface (Figure 11B-D). In general, molecular docking suggests that the registered affinity of **4a** toward the DNA-binding domain of p53 was governed by numerous attractive noncovalent forces, such as hydrogen bonds, electrostatic, π-effects and hydrophobic contacts. In detail, the docking data showed that in the lowest energy binding orientation of **4a**, both the triazole moiety and the 5-FU ring simultaneously formed two hydrogen bonds with the critical residues Arg273 and Glu285 at a distance of 6.78, 4.28 Å, respectively. Furthermore, we also noted that both the chromone and 5-FU moieties in **4a** are involved in three π−cation contacts with the positively charged Arg273. It is also noteworthy that hybrid **4a** establishes three π−alkyl contacts with key Pro250 and Arg248 residues. Besides the key ligand-enzyme interactions described above, diverse hydrophobic interactions were observed during the 4a-binding event, which would have significant importance in stabilizing the **4a**–p53 complex. The observed interactions between promising **4a** and those “hot spot” amino acid residues in p53 allow us to infer that **4a** is able to restore wild-type p53 function in p53 mutant cell lines, which would be in good agreement with experimental measurements.

### 2.4. Molecular Dynamics and Post-MM-PBSA Studies

To verify the docking computational solution obtained for the promising hybrid **4a** against the probable targets (caspase-3, caspase-8 and Tp53), the best-docked pose of **4a** into the each active pocket domain was subjected to molecular dynamic (MD) studies to explore the stability of the ligand-protein complex under physiological conditions (aqueous solution at T = 25 °C, *p* = 1 atm), followed by MM-PBSA studies to estimate the binding free energy. For this purpose, the root-mean-square deviation (RMSD) of atomic positions was examined for all starting complexes after 100 ns MD trajectory. In general, results from the MD simulation showed that **4a** was most stable in caspase-3 than caspase-8 and p53. Thus, as illustrated in Figure 12A, MD data revealed that the complex between caspase-3 and hybrid 4a was very stable throughout the simulation time, without any significant conformational change in the protein structure, reaching the equilibrium within the 100 ns MD simulation trajectories, and showing an average RMSD value of 2.26 ± 0.31 Å, which fall within the optimal range around 2.0 Å [49,50]. In contrast, time evolution RMSD trajectories of compound **4a** in a complex with caspase-8 and p53 showed higher values of RMSD beyond 4Å, meaning large conformational changes in the ligand during the MD simulations (Figure 12B,C).

Furthermore, the radius of gyration (Rg) was also plotted to observe the conformational alterations and dynamic stability of **4a** within each binding domain. The calculated Rg parameter evaluates the compactness of the protein structure and conformational stability of the whole system (i.e., protein-ligand complexes). The lower values of Rg describe a more rigid structure during the simulation. Rg plots for **4a** in complex with caspase-3, caspase-8 and p-53 are illustrated in Figure 12D–F, respectively. Interestingly, a calculated Rg value of 4.72 ± 0.07 Å for hybrid **4a** confirmed the stabilization in the residual backbone and folding of the caspase-3 after the binding event with **4a**. In contrast, the results obtained in the complex with caspase-8 and p53 showed Rg values larger than 5.1 Å. Furthermore, intermolecular H-bond formation between the proteins and **4a** was assessed, and the results are shown in Figure 12G–I. Our findings indicate that **4a** binding to caspase-3 exhibited more H-bond formation at the active site, indicating a greater strength of associations with caspase-3 in comparison to caspase-8 and p53 during the MD simulation period.

These interesting facts were in good agreement with the docking findings, suggesting that the down-modulation of caspase-3 could be the primary biochemical mechanism by which **4a** inhibited the SW620 cells growth and proliferation. With these results in mind, we performed further analysis by MD protocols focused on the complex between hybrid **4a** and caspase-3. Thus, a comparison between the top-scoring binding pose predicted by docking and the equilibrated conformation after 100 ns MD simulation was plotted (Figure 13A), showing no dramatic differences between the structure extracted after MD simulation period and the best docking conformation of **4a**. In addition, the stability of the complex was further assessed by examining the different contributions to the free energy of binding during the formation of the **4a**/caspase-3 complex using molecular mechanics combined with the Poisson–Boltzmann (MM-PBSA) approach. The decomposed contributions of the free energy binding results, which are deposited in Table 3, showed that **4a** binds to caspase-3 enzyme with high affinity (ΔG_bind_ = −22.84 ± 0.08 kcal.mol^−1^). Overall, binding of **4a** and caspase-3 is enhanced by energetic contributions from van der Waals, electrostatic interactions, and the non-polar free energy of solvation (SASA), while a poor contribution of polar solvation energy to the total binding energy is observed after MM-PBSA runs.

The MM-PBSA method was also used to investigate the contribution of individual residues involved in the hybrid **4a**/caspase-3 binding event. Interestingly, the MM-PBSA residue contribution histogram (Figure 13B) revealed that the key surrounding residues on the complex between the compound 4a and caspase-3 resulted to be consistent with those found in docking solutions and MD simulations. Thus, as shown in Figure 13B, the actively involved amino acid residues such as Cys163, Gln161, Ser249, Asp70, His121, Leu194, Ala196, Val239, and Glu246 contributing favorably to the overall binding energy of the **4a**/caspase-3 complex which may have a role in local stabilization of protein complex formation. In contrast, MM-PBSA data showed that four amino acid residues (Arg64, Tyr195, Tyr197, and Val243) contribute negatively to the stability of the ligand-protein complex exhibiting high positive energies. The unfavorable binding contribution of these residues may be strongly associated with bad contacts that create steric clashes, slightly reducing the binding affinity upon complex formation.

## 3. Materials and Methods

### 3.1. Chemical Synthesis

#### 3.1.1. General Remarks

Genistein (synthetic, >95%) and 5-FU (≥98.0%) were purchased from AK scientific chemicals (USA). An ultrasound equipment (BRANSON) was used to assist reactions. NMR spectra were recorded on an AMX 300 instrument (Bruker, Billerica, MA, USA) operating at 300 MHz for ^1^H and 75 MHz for ^13^C. As reference was used the signals of the deuterated solvents and the chemical shifts (δ) were displayed in ppm. TMS was used as an internal standard. Coupling constants (J) are given in Hertz (Hz). HRMS was obtained using a Bruker Impact II UHR-Q-TOF mass spectrometry (Bruker Daltonik GmbH, Bremen Germany) in positive mode. For column chromatography and thin layer chromatography (TLC) silica gel 60 (0.063–0.200 mesh, Merck, Whitehouse Station, NJ, USA) and precoated silica gel plates (Merck 60 F254 0.2 mm) were used.

#### 3.1.2. General Procedure for the Synthesis of Genistein-Bromoalkyl (**1a–h**)

In a flat-bottomed flask of 25 mL equipped with a magnetic stirring bar were placed genistein (1 mmol), DIPEA (1.5 mmol) and DMF (10 mL). The mixture was stirred for a period of 30 min. Then, 1, ω-dibromoalkane (1.2 mmol) was added to the reaction. The mixture was sonicated for a period of 1h to 40 °C. Water was added to the reaction mixture and then was extracted with ethyl acetate. The organic phase was dried on anhydrous sodium sulfate, filtered, concentrated under reduced pressure on a rotatory evaporator and the residue was purified by recrystallization from acetone to obtain bromoalkyl derivatives in yields ranging between 61 and 77%.

7-(2-bromoethoxy)-5-hydroxy-3-(4-hydroxyphenyl)-4H-chromen-4-one (**1a**): Yield 61%; White solid, m.p. 175–180; ^1^H NMR (300 MHz, DMSO-d6) δ 12.98 (OH), 8.41 (s, 1H), 7.39 (d, *J* = 8.6 Hz, 2H), 6.83 (d, *J* = 8.6 Hz, 2H), 6.69 (d, *J* = 2.2 Hz, 1H), 6.43 (d, *J* = 2.2 Hz, 1H), 4.45 (t, *J* = 5.8 Hz, 2H), 3.83 (t, *J* = 5.8 Hz, 2H). ^13^C NMR (75 MHz, DMSO-d6) δ 180.88 (C=O), 164.19, 162.26, 157.94 (O-CH=C, chromone), 157.89, 154.95, 130.63 (2C), 122.99, 121.45, 115.53 (2C), 106.12, 98.86, 93.49, 68.97, 31.40.

7-(3-bromopropoxy)-5-hydroxy-3-(4-hydroxyphenyl)-4H-chromen-4-one (**1b**): Yield 70%; White solid, m.p.124–126; ^1^H NMR (300 MHz, DMSO-d6) δ 8.41 (s, 1H), 7.39 (d, *J* = 8.6 Hz, 2H), 6.83 (d, *J* = 8.6 Hz, 2H), 6.68 (d, *J* = 2.3 Hz, 1H), 6.42 (d, *J* = 2.2 Hz, 1H), 4.20 (t, *J* = 6.0 Hz, 2H), 3.67 (t, *J* = 6.5 Hz, 2H), 2-33-2.21 (m, 2H). ^13^C NMR (75 MHz, DMSO-d6) δ 180.86 (C=O), 164.69, 164.52, 162.21, 157.93 (O-CH=C, chromone), 154.89, 130.62 (2C), 122.95, 121.48, 115.53 (2C), 105.96, 98.84, 93.31, 66.71, 31.99, 31.17.

7-(4-bromobutoxy)-5-hydroxy-3-(4-hydroxyphenyl)-4H-chromen-4-one (**1c**): Yield 74%; White solid, m.p.130–132; ^1^H NMR (300 MHz, DMSO-d6) δ 12.94 (OH), 9.62 (OH), 8.39 (s, 1H), 7.39 (d, *J* = 8.6 Hz, 2H), 6.83 (d, *J* = 8.6 Hz, 2H), 6.63 (d, *J* = 2.2 Hz, 1H), 6.39 (d, *J* = 2.2 Hz, 1H), 4.12 (t, *J* = 6.1 Hz, 1H), 3.61 (t, *J* = 6.5 Hz, 1H), 2.02–1.90 (m, 2H), 1.90–1.77 (m, 2H). ^13^C NMR (75 MHz, DMSO-d6) δ 180.82 (C=O), 164.90, 162.19, 157.91 (O-CH=C, chromone), 154.82, 130.61 (2C), 122.91, 121.50, 115.52 (2C), 105.81, 98.78, 93.27, 68.06, 35.21, 29.36, 27.55.

7-((5-bromopentyl)oxy)-5-hydroxy-3-(4-hydroxyphenyl)-4H-chromen-4-one (**1d**): Yield 75%; White solid, m.p.106–109; ^1^H NMR (300 MHz, DMSO-d6) δ 12.94 (OH), 9.62 (OH), 8.39 (s, 1H), 7.39 (d, *J* = 8.2, 2H), 6.83 (d, *J* = 8.2, 2H), 6.62 (sapparent, 1H), 6.37 (sapparent, 1H), 4.08 (t, *J* = 6.0 Hz, 1H), 3.56 (t, *J* = 6.7 Hz, 1H), 1.94–1.81 (m, 2H), 1.80–1.68 (m, 2H), 1.59–1.45 (m, 2H). ^13^C NMR (75 MHz, DMSO-d6) δ 180.81 (C=O), 165.00, 162.18, 157.92 (O-CH=C, chromone), 154.79, 130.60 (2C), 122.90, 121.50, 115.52 (2C), 105.78, 98.77, 93.22, 68.75, 35.49, 32.34, 27.97, 24.63.

7-((6-bromohexyl)oxy)-5-hydroxy-3-(4-hydroxyphenyl)-4H-chromen-4-one (**1e**): Yield 68%; White solid, m.p.121–125; ^1^H NMR (300 MHz, DMSO-d6) δ 8.39 (s, 1H), 7.39 (d, *J* = 7.9 Hz, 2H), 6.83 (d, *J* = 7.9 Hz, 2H), 6.62 (sapparent, 1H), 6.38 (sapparent, 1H), 4.07 (t, *J* = 5.9 Hz, 2H), 3.54 (t, *J* = 6.5 Hz, 2H), 1.88–1.65 (m, 4H), 1.51–1.34 (m, 4H). ^13^C NMR (75 MHz, DMSO-d6) δ 180.81 (C=O), 165.05, 162.18, 157.92 (O-CH=C, chromone), 154.80, 130.60 (2C), 122.90, 121.50, 115.52 (2C), 105.76, 98.76, 93.21, 68.81, 35.59, 32.61, 28.68, 27.71, 25.02.

7-((8-bromooctyl)oxy)-5-hydroxy-3-(4-hydroxyphenyl)-4H-chromen-4-one (**1f**): Yield 77%; White solid, m.p.115–118; ^1^H NMR (300 MHz, DMSO-d6) δ 8.39 (s, 1H), 7.38 (d, *J* = 8.6 Hz, 2H), 6.82 (d, *J* = 8.6 Hz, 2H), 6.62 (d, *J* = 2.1 Hz, 1H), 6.37 (d, *J* = 2.1 Hz, 1H), 4.06 (t, *J* = 6.4 Hz, 2H), 3.52 (t, *J* = 6.7 Hz, 2H), 1.85–1.63 (m, 4H), 1.48–1.21 (m, 4H). ^13^C NMR (75 MHz, DMSO-d6) δ 180.81 (C=O), 165.07, 162.16, 157.93 (O-CH=C, chromone), 157.89, 154.80, 130.61 (2C), 122.90, 121.50, 115.52 (2C), 105.75, 98.77, 93.21, 68.91, 35.70, 32.66, 29.00, 28.79, 28.50, 27.92, 25.75.

7-((9-bromononyl)oxy)-5-hydroxy-3-(4-hydroxyphenyl)-4H-chromen-4-one (**1g**): Yield 64%; White solid, m.p.120–122; ^1^H NMR (300 MHz, DMSO-d6) δ 12.95 (OH), 9.62 (OH), 8.40 (s, 1H), 7.39 (d, *J* = 8.6 Hz, 2H), 6.82 (d, *J* = 8.6, 2H), 6.63 (sapparent, 1H), 6.38 (sapparent, 1H), 4.07 (d, *J* = 6.0 Hz, 2H), 3.52 (t, *J* = 6.7 Hz, 2H), 1.85–1.65 (m, 4H), 1.45–1.21 (m, 4H). ^13^C NMR (75 MHz, DMSO-d6) δ 180.82 (C=O), 165.08, 162.18, 157.93 (O-CH=C, chromone), 154.82, 130.61 (2C), 122.90, 121.51, 115.52 (2C), 105.75, 98.77, 93.22, 68.92, 35.68, 32.70, 29.28, 29.07, 28.83, 28.51, 27.97, 25.81.

7-((12-bromododecyl)oxy)-5-hydroxy-3-(4-hydroxyphenyl)-4H-chromen-4-one (**1h**): Yield 66%; White solid, m.p.119–122; ^1^H NMR (300 MHz, DMSO-d6) δ 12.95 (OH), 8.36 (s, 1H), 7.37 (d, *J* = 7.8 Hz, 2H), 6.82 (d, *J* = 7.8 Hz, 2H), 6.60 (sapparent, 1H), 6.36 (sapparent, 1H), 4.06 (sapparent, 2H), 3.49 (s sapparent, 2H), 1.86–1.63 (m, 2H), 1.54–1.15 (m, 18H). ^13^C NMR (75 MHz, DMSO-d6) δ 180.82 (C=O), 165.09, 162.23, 159.28 (O-CH=C, chromone), 154.66, 130.54 (2C), 122.99, 121.48, 115.51 (2C), 105.80, 98.73, 93.13, 68.86, 35.43, 32.71, 29.40 (4C), 29.17, 28.84, 28.60, 27.99.

#### 3.1.3. General Procedure for the Synthesis of Genistein-Alkylazides (**2a–h**)

In a 10 mL flat-bottomed flask were mixed compounds **1a**–**h** (1 mmol), sodium azide (3 mmol) and 5mL of DMF. This mixture was sonicated for a period of 1h to 40 °C. After this time, water was added and extracted with hexane. Then, anhydrous sodium sulfate was added to the organic phase and the heterogeneous solution was filtered; the liquid obtained was concentrated under reduced pressure on a rotatory evaporator to obtain compounds 2a-h in yields ranging between 78 and 89%.

7-(2-azidoethoxy)-5-hydroxy-3-(4-hydroxyphenyl)-4H-chromen-4-one (**2a**): Yield 83%; White solid, m.p.123–125; ^1^H NMR (300 MHz, DMSO-d6) δ 8.40 (s, 1H), 7.39 (d, *J* = 8.4 Hz, 2H), 6.83 (d, *J* = 8.4 Hz, 2H), 6.67 (d, *J* = 1.8 Hz, 1H), 6.41 (d, *J* = 1.8 Hz, 1H), 4.29 (t, *J* = 4.6, 2H), 3.70 (t, *J* = 4.6, 2H),2.55–2.47 (m, 2H). ^13^C NMR (75 MHz, DMSO-d6) δ 180.87 (C=O), 164.25, 162.24, 157.94 (O-CH=C, chromone), 157.88, 154.92, 130.62 (2C), 123.00, 121.43, 115.54 (2C), 106.10, 98.78, 93.37, 68.01, 49.76, 36.25.

7-(3-azidopropoxy)-5-hydroxy-3-(4-hydroxyphenyl)-4H-chromen-4-one (**2b**): Yield 83%; White solid, m.p.124–127; ^1^H NMR (300 MHz, DMSO-d6) δ 12.96 (s, OH), 8.41 (s, OH), 7.95 (s, 1H), 7.39 (d, *J* = 8.6 Hz, 2H), 6.83 (d, *J* = 8.6 Hz, 2H), 6.67 (d, *J* = 2.2 Hz, 1H), 6.41 (d, *J* = 2.2 Hz, 1H), 4.16 (t, *J* = 6.1 Hz, 2H), 3.53 (d, *J* = 6.7 Hz, 1H), 2.54–2.45 (m, 2H). ^13^C NMR (75 MHz, DMSO-d6) δ 180.85 (C=O), 164.74, 162.76, 162.19, 157.93 (O-CH=C, chromone), 154.88, 130.62 (2C), 122.95, 121.48, 115.53 (2C), 105.92, 98.84, 93.29, 66.11, 55.38, 47.99, 36.24, 31.21, 28.32.

7-(4-azidobutoxy)-5-hydroxy-3-(4-hydroxyphenyl)-4H-chromen-4-one (**2c**): Yield 88%; White solid, m.p.123–125. ^1^H NMR (300 MHz, DMSO-d6) δ 12.94 (OH), 9.64 (OH), 8.38 (s, 1H), 7.38 (d, *J* = 8.6 Hz, 2H), 6.83 (d, *J* = 8.6 Hz, 2H), 6.61 (d, *J* = 2.2 Hz, 1H), 6.37 (d, *J* = 2.2 Hz, 1H), 4.09 (t, *J* = 6.1 Hz, 2H), 3.41 (t, *J* = 6.7 Hz, 2H), 1-84-1.61 (m, 4H). ^13^C NMR (75 MHz, DMSO-d6) δ 180.80 (C=O), 164.90, 162.18, 157.93 (O-CH=C, chromone), 157.89, 154.77, 130.59 (2C), 122.90, 121.50, 115.52 (2C), 105.79, 98.75, 93.21, 68.37, 50.76, 26.09, 25.43.

7-((5-azidopentyl)oxy)-5-hydroxy-3-(4-hydroxyphenyl)-4H-chromen-4-one (**2d**): Yield 81%; White solid, m.p.74–78. ^1^H NMR (300 MHz, DMSO-d6) δ 12.94 (OH), 9.63 (OH), 8.38 (s, 1H), 7.39 (d, *J* = 8.6 Hz, 2H), 6.83 (d, *J* = 8.6 Hz, 2H), 6.62 (d, *J* = 2.2 Hz, 2H), 6.37 (d, *J* = 2.2 Hz, 2H), 4.07 (t, *J* = 6.4 Hz, 2H), 3.35 (t, *J* = 6.8 Hz, 2H), 1-81-1.68 (m, 2H), 1-66-1.53 (m, 2H), 1-52-1.39 (m, 2H). ^13^C NMR (75 MHz, DMSO-d6) δ 180.80 (C=O), 165.01, 162.18, 157.92 (O-CH=C, chromone), 154.78, 130.60 (2C), 122.90, 121.50, 115.51 (2C), 105.77, 98.76, 93.21, 68.75, 51.00, 28.42, 28.35, 23.17.

7-((6-azidohexyl)oxy)-5-hydroxy-3-(4-hydroxyphenyl)-4H-chromen-4-one (**2e**): Yield 84%; White solid, m.p.93–97. ^1^H NMR (300 MHz, DMSO-d6) δ 12.94 (OH), 9.62 (OH), 8.37 (s, 1H), 7.38 (d, *J* = 8.5 Hz, 2H), 6.83 (d, *J* = 8.6 Hz, 2H), 6.59 (d, *J* = 2.1 Hz, 1H), 6.36 (d, *J* = 2.1 Hz, 1H), 4.05 (t, *J* = 6.4 Hz, 2H), 3.32 (t, *J* = 6.8 Hz, 2H), 1.77–1.65 (m, 2H), 1.60–1.49 (m, 2H), 1.47–1.30 (m, 4H). ^13^C NMR (75 MHz, DMSO-d6) δ 180.79 (C=O), 165.03, 162.18, 157.92 (O-CH=C, chromone), 157.90, 154.74, 130.58 (2C), 122.89, 121.51, 115.51 (2C), 105.75, 98.73, 93.16, 68.80, 51.02, 28.72, 28.64, 26.33, 25.42.

7-((8-azidooctyl)oxy)-5-hydroxy-3-(4-hydroxyphenyl)-4H-chromen-4-one (**2f**): Yield 89%; White solid, m.p.113–117. ^1^H NMR (300 MHz, DMSO-d6) δ 12.94 (OH), 9.62 (OH), 8.37 (s, 1H), 7.38 (d, *J* = 8.6 Hz, 2H), 6.82 (d, *J* = 8.6 Hz, 2H), 6.59 (d, *J* = 2.2 Hz, 1H), 6.35 (d, *J* = 2.2 Hz, 1H), 4.04 (t, *J* = 6.4 Hz, 2H), 3.30 (t, *J* = 6.8 Hz, 2H), 1.77–1.63 (m, 2H), 1.58–1.44 (m, 2H), 1.45–1.24 (m, 4H). ^13^C NMR (75 MHz, DMSO-d6) δ 180.79 (C=O), 165.06, 162.18, 157.92 (O-CH=C, chromone), 154.74, 130.58 (2C), 122.89, 121.50, 115.51 (2C), 105.74, 98.72, 93.15, 68.89, 51.06, 29.06, 28.94, 28.81, 28.68, 26.55, 25.76.

7-((9-azidononyl)oxy)-5-hydroxy-3-(4-hydroxyphenyl)-4H-chromen-4-one (**2g**): Yield 85%; White solid, m.p.109–111. ^1^H NMR (300 MHz, DMSO-d6) δ 12.94 (OH), 9.62 (OH), 8.37 (s, 1H), 7.38 (d, *J* = 8.6 Hz, 2H), 6.82 (d, *J* = 8.6 Hz, 2H), 6.59 (d, *J* = 2.1 Hz, 1H), 6.35 (d, *J* = 2.1 Hz, 1H), 4.04 (t, *J* = 6.4 Hz, 2H), 3.29 (t, *J* = 6.8 Hz, 2H), 1.76–1.63 (m, 2H), 1.56–1.44 (m, 2H), 1.44–1.20 (m, 10H). ^13^C NMR (75 MHz, DMSO-d6) δ 180.78 (C=O), 165.05, 162.19, 157.91 (O-CH=C, chromone), 154.73, 130.57 (2C), 122.89, 121.50, 115.51 (2C), 105.74, 98.73, 93.14, 68.89, 51.06, 29.32, 29.10, 28.95, 28.84, 28.70, 26.59, 25.82.

7-((12-azidododecyl)oxy)-5-hydroxy-3-(4-hydroxyphenyl)-4H-chromen-4-one (**2h**): Yield 82%; White solid, m.p.115–117; ^1^H NMR (300 MHz, DMSO-d6) δ 12.93 (s, OH), 9.58 (s, OH), 8.36 (s, 1H), 7.37 (d, *J* = 8.1 Hz, 2H), 6.82 (d, *J* = 8.1 Hz, 2H), 6.58 (sapparent, 1H), 6.35 (sapparent, 1H), 4.05 (sapparent, 2H), 3.28 (sapparent, 2H), 1.82–1.63 (m, 2H), 1.59–1.45 (m, 2H), 1.44–1.13 (m, 16H). ^13^C NMR (75 MHz, DMSO-d6) δ 180.81 (C=O), 165.07, 162.24, 157.94 (O-CH=C, chromone), 154.62, 130.53 (2C), 122.98, 121.48, 115.50 (2C), 105.79, 98.72, 93.10, 68.86, 51.08, 29.41 (4C), 29.18, 29.03, 28.85, 28.72, 26.61, 25.84.

#### 3.1.4. Syntheses of Propargyl-5-FU (3): 

5-fluorouracil (1 mmol), propargyl bromide (1.1 mmol), K_2_CO_3_ (1.1 mmol) and DMF (2 mL) were placed in a 10 mL flat-bottomed flask. For 1 h and at 40 °C, the mixture was placed in the ultrasonic bath. Subsequently, HCl 10% (5 mL) was added and extracted with ethyl acetate. Anhydrous sodium sulfate was used to dry the organic phase, which, after filtration, was concentrated on a rotatory evaporator and the residue purified by column chromatography over silica gel using a mixture of hexane/ethyl acetate of different ratios to obtain compounds 3 in 40% yield.

#### 3.1.5. General Procedure for the Synthesis of 5-FU-Genistein Hybrids (**4a–h**)

A mixture of propargyl-5-FU (3) (1 mmol), Genistein-alkylazides (**2a–h**) (1 mmol) and DMF (5 mL) were placed in a 10 mL flat-bottomed flask and sonicated for 5 min to 40 °C. Subsequently, a mixture of ascorbic acid (0.5 mmol), copper acetate (0.5 mmol), DMF (1mL) and water (1 mL) was added and sonicated for 1 h to 40 °C. HCl 10% was added, and reaction mixture was then extracted with ethyl acetate. Anhydrous sodium sulfate was used to dry the organic phase, which, after filtration, was concentrated on a rotatory evaporator and the residue purified by crystallization (MeOH:H_2_O, 1:1 ratio). Column chromatography over silica gel using a mixture of hexane/ethyl acetate of different ratios, to obtain compounds 3 in 40% yield. Preparative chromatography over silica gel eluting with a mixture of hexane/ethyl acetate of different ratios was used to purify the solid obtained. Compounds **4a–h** in yields ranging between 43 and 94%. The ^1^H, ^13^C NMR and MS spectra (Appendix A) and HPLC analysis (Appendix A) of all hybrids can be found in the Appendix A.

5-fluoro-1-((1-(3-((5-hydroxy-3-(4-hydroxyphenyl)-4-oxo-4H-chromen-7-yl)oxy)ethyl)-1H-1,2,3-triazol-4-yl)methyl)pyrimidine-2,4(1H,3H)-dione (**4a**): Yield 61%; White solid, m.p. 161–164; ^1^H NMR (600 MHz, DMSO-d6) δ 9.60 (s, 1H), 8.39 (s, 1H), 8.17 (s, 1H), 7.40 (d, *J* = 6.7 Hz, 2H), 6.83 (d, *J* = 6.7 Hz, 2H), 6.60 (s, 1H), 6.38 (s, 1H), 4.91 (s, 2H), 4.53 (sapparent, 2H), 4.12 (sapparent, 2H), 2.35–2.26 (m, 2H). ^13^C NMR (151 MHz, DMSO-d6) δ 180.86 (C=O chromone), 164.03 (Ar-O), 162.22 (Ar-O), 157.99 and 157.93 (F-C-C=O), 157.84 (Ar-O), 157.74 (Ar-O), 154.93 (-O-CH=C-,chromone), 149.84 (N-C=O), 142.87 (triazolyl),141.63 and 138.59 (F-C), 130.61 (2C-Ar), 130.11, 130.25 and 130.12 (CH-C-F), 124.82 (triazolyl), 122.99, 121.44, 115.53 (2C-Ar), 106.14, 98.85, 93.45, 67.24, 49.28, 43.16. HRMS-ESI (*m/z*): 508.1293 [M+H]^+^ calcd. for C_24_H_19_FN_5_O_7_ [M + H]^+^ 508.1263.

5-fluoro-1-((1-(3-((5-hydroxy-3-(4-hydroxyphenyl)-4-oxo-4H-chromen-7-yl)oxy)propyl)-1H-1,2,3-triazol-4-yl)methyl)pyrimidine-2,4(1H,3H)-dione (**4b**): Yield 82%; White solid, m.p. 175–178; ^1^H NMR (600 MHz, DMSO-d6) δ 8.40 (s, 1H), 8.20 (s, 1H), 8.14 (d, *J* = 6.6 Hz, 1H), 7.39 (d, *J* = 8.5 Hz, 2H), 6.83 (d, *J* = 8.5 Hz, 2H), 6.65 (s, 1H), 6.39 (s, 1H), 4.91 (s, 2H), 4.79 (t, *J* = 5.0 Hz, 2H), 4.54 (t, *J* = 4.7 Hz, 2H). ^13^C NMR (151 MHz, DMSO-d6) δ 180.87 (C=O chromone), 164.65 (Ar-O), 162.22 (Ar-O), 157.99 and 157.94 (F-C-C=O), 157.90 (Ar-O), 157.79 (Ar-O), 154.85 (-O-CH=C-,chromone), 149.84 (N-C=O), 142.87 (triazolyl), 140.89 and 139.37 (F-C), 130.62 (3C-Ar), 130.43 and 130.21 (CH-C-F), 124.30 (triazolyl), 122.97, 121.51, 115.55 (2C-Ar), 105.96, 98.80, 93.31, 66.06, 47.05, 43.21, 29.54. HRMS-ESI (*m/z*): 522.1447 [M+H]^+^ calcd. for C_25_H_21_FN_5_O_7_ [M + H]^+^ 522.1419.

5-fluoro-1-((1-(3-((5-hydroxy-3-(4-hydroxyphenyl)-4-oxo-4H-chromen-7-yl)oxy)butyl)-1H-1,2,3-triazol-4-yl)methyl)pyrimidine-2,4(1H,3H)-dione (**4c**): Yield 84%; White solid, m.p. 200–203; ^1^H NMR (600 MHz, DMSO-d6) δ 8.40 (s, 1H), 8.18 (d, *J* = 6-6 Hz, 1H), 8.14 (s, 1H), 7.40 (d, *J* = 8.2 Hz, 2H), 6.83 (d, *J* = 8.2, 2H), 6.64 (s, 1H), 6.40 (s, 1H), 4.91 (s, 2H), 4.43 (t, *J* = 6.8 Hz, 2H), 4.11 (t, *J* = 5.9 Hz, 2H), 2.01–1.94 (m, 2H), 1.75–1.68 (m, 2H). ^13^C NMR (151 MHz, DMSO-d6) δ 180.85 (C=O chromone), 164.91 (Ar-O), 162.22 (Ar-O), 157.99 (F-C-C=O), 157.94 (Ar-O), 157.82 (Ar-O), 154.82 (-O-CH=C-,chromone), 149.84 (N-C=O), 142.60 (triazolyl), 140.89 and 139.37 (F-C), 130.61 (2C-Ar), 130.50 and 130.28 (CH-C-F), 124.02 (triazolyl), 122.96, 121.53, 115.55 (2C-Ar), 105.86, 98.80, 93.30, 68.26, 49.52, 43.24, 26.83, 25.88. HRMS-ESI (*m/z*): 536.1601 [M+H]^+^ calcd. for C_26_H_23_FN_5_O_7_ [M + H]^+^ 536.1576.

5-fluoro-1-((1-(3-((5-hydroxy-3-(4-hydroxyphenyl)-4-oxo-4H-chromen-7-yl)oxy)pentyl)-1H-1,2,3-triazol-4-yl)methyl)pyrimidine-2,4(1H,3H)-dione (**4d**): Yield 70%; White solid, m.p. 211–214; ^1^H NMR (600 MHz, DMSO-d6) δ 9.60 (s, 1H), 8.39 (s, 1H), 8.18 (d, *J* = 6.4 Hz, 1H), 8.12 (s, 1H), 7.40 (d, *J* = 8.2 Hz, 2H), 6.83 (d, *J* = 8.2 Hz, 2H), 6.63 (s, 1H), 6.38 (s, 1H), 4.90 (s, 2H), 4.37 (t, *J* = 6.8 Hz, 3H), 4.08 (t, *J* = 5.9 Hz, 3H), 1.92–1.85 (m, 2H), 1.79–1.73 (m, 2H), 1.43–1.36 (m, 2H). ^13^C NMR (151 MHz, DMSO-d6) δ 180.84 (C=O chromone), 165.02 (Ar-O), 162.21 (Ar-O), 157.99 (F-C-C=O), 157.94 (Ar-O), 157.82 (Ar-O), 154.81 (-O-CH=C-,chromone), 149.84 (N-C=O), 142.58 (triazolyl), 140.88 and 139.36 (F-C), 130.62 (3C-Ar), 130.49 and 130.27 (CH-C-F), 123.94 (triazolyl), 122.95, 121.53, 115.54 (2C-Ar), 105.82, 98.79, 93.28, 68.70, 49.77, 43.23, 29.77, 28.18, 22.88. HRMS-ESI (*m/z*): 550.1756 [M+H]^+^ calcd. for C_27_H_25_FN_5_O_7_ [M+H]^+^ 550.1733.

5-fluoro-1-((1-(3-((5-hydroxy-3-(4-hydroxyphenyl)-4-oxo-4H-chromen-7-yl)oxy)hexyl)-1H-1,2,3-triazol-4-yl)methyl)pyrimidine-2,4(1H,3H)-dione (**4e**): Yield 82%; White solid, m.p. 220–223; ^1^H NMR (600 MHz, DMSO-d6) δ 8.39 (s, 1H), 8.17 (d, *J* = 6.3 Hz, 1), 8.12 (s, 1H), 7.40 (d, *J* = 8.0 Hz, 2H), 6.83 (d, *J* = 8.0 Hz, 2H), 6.63 (s, 1H), 6.39 (s, 1H), 4.90 (s, 2H), 4.35 (t, *J* = 6.6 Hz, 2H), 4.07 (t, *J* = 5.8 Hz, 2H), 1.88–1.80 (m, 2H), 1.76–1.67 (m, 2H), 1.48–1.39 (m, 2H), 1.34–1.27 (m, 2H). ^13^C NMR (151 MHz, DMSO-d6) δ 180.84 (C=O chromone), 165.07 (Ar-O), 162.21 (Ar-O), 157.99 (F-C-C=O), 157.95 (Ar-O), 157.82 (Ar-O), 154.80 (-O-CH=C-,chromone), 149.83 (N-C=O), 142.58 (triazolyl), 140.88 and 139.36 (F-C), 130.61 (3C-Ar), 130.49 and 130.27 (CH-C-F), 123.90 (triazolyl), 122.95, 121.53, 115.54 (2C-Ar), 105.80, 98.79, 93.25, 68.83, 49.81, 43.24, 30.00, 28.63, 26.01, 25.26. HRMS-ESI (*m/z*): 564.1907 [M+H]^+^ calcd. for C_28_H_27_FN_5_O_7_ [M+H]^+^ 564.1889.

5-fluoro-1-((1-(3-((5-hydroxy-3-(4-hydroxyphenyl)-4-oxo-4H-chromen-7-yl)oxy)octyl)-1H-1,2,3-triazol-4-yl)methyl)pyrimidine-2,4(1H,3H)-dione (**4f**): Yield 60%; White solid, m.p. 212–215; ^1^H NMR (600 MHz, DMSO-d6) δ 8.39 (s, 1H), 8.17 (d, *J* = 6.5 Hz, 1H), 8.11 (s, 1H), 7.39 (d, *J* = 8.3 Hz, 2H), 6.83 (d, *J* = 8.3 Hz, 2H), 6.63 (s, 1H), 6.39 (s, 1H), 4.90 (s, 2H), 4.33 (t, *J* = 7.0 Hz, 2H), 4.07 (t, *J* = 6.3 Hz, 1H), 1.84–1.77 (m, 2H), 1.75–1.67 (m, 2H), 1.42–1.35 (m, 2H), 1.34–1.27 (m, 4H), 1.27–1.20 (m, 2H). ^13^C NMR (151 MHz, DMSO-d6) δ 180.84 (C=O chromone), 165.11 (Ar-O), 162.20 (Ar-O), 157.97 (Ar-O), 157.93 (F-C-C=O), 157.82 (Ar-O), 154.80 (-O-CH=C-,chromone), 149.82 (N-C=O), 142.75 (triazolyl), 140.88 and 139.35 (F-C), 130.61 (3C-Ar), 130.49, 130.27, 123.99 (triazolyl), 122.94, 121.53, 115.54 (2C-Ar), 105.79, 98.79, 93.25, 68.93, 49.87, 43.24, 30.06, 28.97, 28.79, 28.74, 26.24, 25.74. HRMS-ESI (*m/z*): 592.2216 [M+H]^+^ calcd. for C_30_H_31_FN_5_O_7_ [M+H]^+^ 592.2202.

5-fluoro-1-((1-(3-((5-hydroxy-3-(4-hydroxyphenyl)-4-oxo-4H-chromen-7-yl)oxy)nonyl)-1H-1,2,3-triazol-4-yl)methyl)pyrimidine-2,4(1H,3H)-dione (**4g**): Yield 84%; White solid, m.p. 191–194; ^1^H NMR (600 MHz, DMSO-d6) δ 9.60 (s, 1H), 8.39 (s, 1H), 8.17 (d, *J* = 5.5 Hz, 1H), 8.11 (s, 1H), 7.39 (d, *J* = 7.5 Hz, 2H), 6.83 (d, *J* = 7.7 Hz, 2H), 6.63 (s, 1H), 6.39 (s, 1H), 4.89 (s, 2H), 4.32 (sapparent, 2H), 4.07 (sapparent, 2H), 1.85–1.76 (m, 2H), 1.75–1.68 (m, 2H), 1.45–1.18 (m, 10H). ^13^C NMR (151 MHz, DMSO-d6) δ 180.83 (C=O chromone), 165.11(Ar-O), 162.20 (Ar-O), 157.95 (F-C-C=O and Ar-O), 157.81 (Ar-O), 154.77 (-O-CH=C-,chromone), 149.82 (N-C=O), 143.25 (triazolyl), 140.88 and 139.35 (F-C), 130.60 (3C-Ar), 130.47, 130.25, 124.18 (triazolyl), 122.94, 121.53, 115.54 (2C-Ar), 105.78, 98.78, 93.23, 68.93, 49.88, 43.27, 30.05, 29.22, 29.05, 28.82, 28.73, 26.26, 25.79. HRMS-ESI (*m/z*): 606.2370 [M+H]^+^ calcd. for C_31_H_33_FN_5_O_7_ [M+H]^+^ 606.2358.

5-fluoro-1-((1-(3-((5-hydroxy-3-(4-hydroxyphenyl)-4-oxo-4H-chromen-7-yl)oxy)dodecyl)-1H-1,2,3-triazol-4-yl)methyl)pyrimidine-2,4(1H,3H)-dione (**4h**): Yield 80%; White solid, m.p. 190–193; ^1^H NMR (600 MHz, DMSO-d6) δ 8.39 (s, 1H), 8.16 (d, *J* = 6.63, 1H), 8.10 (s, 1H), 7.39 (d, *J* = 8.4 Hz, 2H), 6.83 (d, *J* = 8.4 Hz, 2H), 6.63 (d, *J* = 1.8 Hz, 1H), 6.38 (d, *J* = 1.8 Hz, 1H), 4.89 (s, 2H), 4.31 (t, *J* = 7.1 Hz, 2H), 4.08 (t, *J* = 6.4 Hz, 2H), 1.81–1.75 (m, 2H), 1.75–1.69 (m, 2H), 1.43–1.36 (m, 2H), 1.34–1.18 (m, 14H). ^13^C NMR (151 MHz, DMSO-d6) δ 180.83 (C=O chromone), 165.11 (Ar-O), 162.21 (Ar-O), 157.95 (F-C-C=O and Ar-O), 157.80 (Ar-O), 154.78 (-O-CH=C-,chromone), 149.82 (N-C=O), 142.59 (triazolyl), 140.88 and 139.35 (F-C), 130.59 (3C-Ar), 130.46, 130.24, 123.92 (triazolyl), 122.94, 121.51, 115.54 (2C-Ar), 105.78, 98.78, 93.23, 68.94, 49.86, 43.20, 34.71, 30.08, 29.41, 29.37, 29.30, 29.15, 28.84, 28.81, 26.28, 25.83. HRMS-ESI (*m/z*): 648.2833 [M+H]^+^ calcd. for C_34_H_39_FN_5_O_7_ [M+H]^+^ 648.2828.

### 3.2. In Vitro Biological Assays

#### 3.2.1. Cell Lines and Culture Conditions

Two colon cancer cells (SW480 and SW620) and non-malignant cell lines (human keratinocytes, HaCaT; chinese hamster ovary, CHO-K1) were obtained from The European Collection of Authenticated Cell Cultures (ECACC, England). For biological evaluations, cells were cultured in Dulbecco’s Modified Eagle Medium, supplemented with 10% heat-inactivated (56 °C) horse serum, 1% penicillin/streptomycin and 1% non-essential amino acids (Gibco Invitrogen, Carlsbad, CA, USA). During the treatments, the serum was reduced to 3%, and the medium was supplemented with insulin (10 mg/mL) transferrin (5 mg/mL) and selenium (5 ng/mL) (ITS-defined medium; Gibco, Invitrogen, Carlsbad, USA) [51,52]. To avoid *Mycoplasma* spp. contamination, cell cultures were regularly tested by polymerase chain reaction.

#### 3.2.2. Cytotoxic and Antiproliferative Activity

The hybrid compounds evaluated together with the parental compounds, the reference drug (5-FU) and the equimolar mixture (this was obtained by mixing 6.03 mg of genistein and 2.90 mg of 5-FU in 1 mL of solvent) were evaluated through the Sulforhodamine B (SRB) assay to demonstrate the probable cytotoxic and antiproliferative potential. Briefly, cells were seeded at 20,000 cells/well (for cytotoxicity assay) and 2500 cells/well (for the antiproliferative test) in 96-well tissue culture plates, incubating at 37 °C in a humidified atmosphere with 5% CO_2_. After the adaptation time (24 h), cells were treated with increasing concentrations of the hybrid molecules or DMSO (1%). At the end of each treatment, cells were fixed with trichloroacetic acid (50% *v*/*v*; MERCK) for one hour at 4 °C. Subsequently, cell proteins were stained with SRB (0.4% *w*/*v*; Sigma-Aldrich, United States) and washed with acetic acid (1%). After drying at room temperature, Tris-base (10 mM) was added to solubilize protein bound stain. Absorbance was read in a microplate reader (Mindray MR-96A) at 492 nm [53]. All experiments were performed at least in duplicate.

#### 3.2.3. Effect of 5-FU/Genistein Hybrids on Cell Cycle Distribution

Cell cycle analysis was evaluated by flow cytometry using propidium iodide (PI). This experiment was carried out as previously described by Nicoletti et al. (1991) [54]. Then, 48 h after treatment with the IC_50_ value of the hybrids **4a**, **4g** and DMSO (1%) as a vehicle control, cells were collected by scraping and centrifugation, resuspending the cell pellet in a versene buffer. Thereafter, cells were fixed with 1.8 mL cold ethanol (70%) at 4 °C overnight. Alcohol was removed by subsequent washing twice with versene buffer. The final pellet was resuspended in 300 µL of PBS with 0.25 mg/mL RNAse (Type I-A, Sigma-Aldrich, Germany) and 0.1 mg/mL PI and incubated for 30 min in the dark at room temperature. FACS Canto II flow cytometer (BD Biosciences, USA) was used to read the PI fluorescence of 10,000 events. The analysis of data was performed with the software FlowJo 7.6.2 (Ashland, OR, USA) [55].

#### 3.2.4. Measurement of Mitochondrial Membrane Potential (ΔΨm)

The fluorescent dyes DiOC6 (3,3′-dihexyloxacarbocyanine iodide, Thermo Fisher Scientific, Waltham, MA, USA) and propidium iodide (PI) were used to evaluate possible changes in mitochondrial membrane potential induced by the hybrid molecules. Then, 2.5 × 10^5^ cells/well were seeded in 6-well tissue culture plates and allowed to grow for 24 h. Then, these were treated with hybrids **4a** and **4g** (with the IC_50_ value for each compound). Cells were harvested by scrapping 48 h after treatment, stained with DiOC6 and propidium iodide (PI) at room temperature for 30 min in darkness. Flow cytometric analysis of 10,000 events was carried out to quantify cells with mitochondrial depolarization, excitation at 488 nm and detection of the emission with the green (530/15 nm) and the red (610/20 nm) filters [55].

#### 3.2.5. Cell Death Induction by Hybrids **4a** and **4g**

Using double staining with Annexin V/FITC and PI (Roche Diagnostics), we evaluated the ability of the hybrids **4a** and **4g** to induce cell death. SW480 and SW620 cells were treated for 48 h and further collected by scraping. After centrifugation, the cell pellet was resuspended in Annexin V/FITC–PI and incubated in darkness for 20 min. Data were obtained through flow cytometry and analyzed with FlowJo 7.6.2 (Ashland, OR, USA). Cells in early apoptosis showed single staining for Annexin V/FITC. Cells with positive staining for PI were considered dead cells, late apoptotic cells, necroptotic cells or secondary necrotic cells. All assays were repeated twice [56].

#### 3.2.6. Determination of Apoptotic Biomarkers

Tumor cell lines (SW480 and SW620) were cultured as previously described and then treated with hybrids **4a** (62.73 and 50.58 µM, for SW480 and SW620 cells, respectively) and **4g** (over SW620, IC_50_ = 36.84 µM) for 48 h. Afterwards, cells were scraped and lysed with Cell Lysis Buffer (1X, Ref. #9803). The supernatant was used for the determination of apoptotic biomarkers after the treatment with the hybrid molecules. Enzyme-linked immunosorbent assay (ELISA) kits were provided by Elabscience Biotechnology Co., (Wuhan, Hubei, China) (caspase-8) and Cell-Signaling Technology (Danvers, MA, USA) (caspase-3 and p53), following the manufacturer’s instructions [57].

#### 3.2.7. Statistical Analysis

The experimental design included at least two repetitions for each experiment. Data were reported as mean ± SE (standard error). Statistical analysis was performed by one-way ANOVA followed by Dunnett’s test. *p* values lower than 0.05 were considered significant. Data were analyzed with GraphPad Prism version 8.0.1 for (Graph Pad Software, San Diego, CA, USA).

### 3.3. Computational Methods

#### 3.3.1. Docking Protocol

The 2D chemical structure of **4a** was drawn in the ChemDraw 17.0 software (Cambridge Soft, USA) and then saved as MDL MoL files. Chem3D 17.0 software (Cambridge Soft, USA) was used to generate the 3D structure of the ligand and optimization was performed using the MM2 Force-Field in Chem3D Ultra 8.0 Software CS, ChemOffice Chem3D Ultra 8.0, CambridgeSoft, AutodockTools (ADT) was used to parameterize the ligand: non-polar hydrogens were merged, rotatable bonds assigned, full hydrogens added, and Kollman united partial atom charges were added to the individual protein atoms. The 3D protein structure of the Caspase-3 (PDB ID: 5I9B), Caspase-8 (PDB code: 3kjn) and p53 (PDB ID: 1tsr) was downloaded from the Protein Data Bank website (accessed on 1 June 2022). Co-crystallized ligand, ions and water molecules were removed from the protein structure by using the DS Visualizer 2.5 program. For the docking analysis, grid map dimensions (32 × 32 × 32 Å) were set surrounding the active site at x, y, and z coordinates of x = 1.5, y = −8.1, z = −13.4 (for caspase-3), x = −10.032, y = 35.318, z = 41.367 (for caspase-8) and x = 56.907, y = 28.404, z = 85.947 (for p53), at exhaustiveness 20 for each protein-compound pair and 1Å for grid spacing. AutoDock Vina software package [54] was used, adopting a flexible-ligand/rigid-receptor protocol and binding affinity/free energy estimated in kcal/mol. Finally, to inspect docking solutions, DS Visualizer 2.5 and PyMOL Molecular Graphics System Version 2.0 Schrodinger, LLC (2015) were used.

#### 3.3.2. Molecular Dynamics (MD) Studies

The protocol employed and procedural details for MD simulations (100 ns) have been described in a recent publication [58].

#### 3.3.3. Binding Energy Calculation (MM/PBSA Protocol)

The protocols employed for the molecular mechanics Poisson–Boltzmann surface area (MM/PBSA) calculations have been described in a recent publication [58].

## 4. Conclusions

We showed for the first time the synthesis and biological evaluation of 5-FU/Genistein hybrids. We demonstrated that the synthesized hybrids exhibited selective chemopreventive potential towards the human colon adenocarcinoma cells SW480 and SW620. The most active hybrids **4a** and **4g** displayed cytotoxicity against SW480 and SW620 cells with IC_50_ values between 36.84 ± 0.71 and 62.73 ± 7.26 μM. These compounds were even more selective than genistein alone, the reference drug (5-FU) and the equimolar mixture. Furthermore, we also demonstrated that hybrids **4a** and **4g** produced time- and concentration-dependent antiproliferative activity and cell cycle arrest at the S-phase and G2/M, inducing morphological changes in the cell. Our results allowed us to conclude that the probable mechanism of action of hybrid **4a** in SW620 cells could probably be triggered by extrinsic apoptosis in response to the activation of Tp53. Finally, the molecular docking analysis showed that promising **4a** bound effectively to caspases-3/7 and Tp53 protein obtaining binding affinities ranging from −7.9 to −9.7 kcal/mol compared to those current inhibitors. MD simulations not only validated the stability and rationality of docking solutions, but also showed that the complex between the hybrid **4a** and caspase-3 achieved great stability in 100 ns. In addition, the MM-PBSA-based analysis showed that compound **4a** binds to the caspase-3 enzyme with high affinity (ΔG_bind_ = −22.84 ± 0.08 kcal.mol^−1^). Therefore, combined experimental and computational findings indicated that the modulation of these proteins, particularly caspase-3, may be a possible molecular mechanism to understand the cytotoxic response of **4a** in SW620 colon carcinoma cell lines. However, further experimental and computational studies are needed to clearly delineate the cytotoxic mechanism associated with the most promising hybrid (**4a)**.

## Data Availability

Data is contained within the article.

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
