# Peer review of "Synthesis and Chemopreventive Potential of 5-FU/Genistein Hybrids on Colorectal Cancer Cells"

_pharmaceuticals, 2022, doi:10.3390/ph15101299_

Round 1
Reviewer 1 Report
Dear editor,
Have a nice day. The authors synthesized a new series of 5-FU-Genistein hybrids. The synthesized compounds were evaluated in vitro for their cytotoxic activities against tumor and normal cell lines. The apoptotic effect of the most active member was assessed. In silico docking studies were carried out.
This wok needs some minor modifications prior its publication as follows.
1- The chemical name should be started with capital letters.
2- The J symbol in chemistry section should be italic.
3- The NMR charts should be provided in supp data.
4- The purity of compounds should be assessed using HPLC method or HRMS.
5- The number 1 and 13 in NMR should be superscript.
6- Docking of the reference molecule should be carried out. then compare the binding mode of the tested compounds with that of the reference molecule.
7- Validation of the docking studies should be assessed using MD simulations at 100 ns.
Author Response
Dear Editor,
As regards our manuscript entitled "Synthesis and Chemopreventive Potential of 5-FU/Genistein Hybrids on Colorectal Cancer Cells" by Gustavo Moreno-Q et al., sent for publication in Pharmaceuticals. We inform you that the manuscript has been thoroughly corrected according to the comments made by the reviewer.
We hope that the corrections we have introduced in the manuscript serve to meet the high standards of quality of Pharmaceuticals and that you will consider it for publication.
Yours sincerely
______________________
Wilson Cardona-G
Chemistry Institute
Universidad de Antioquia
Colombia

Reviewer 2 Report
The authors Moreno-Q, Castrillon-L and co-workers submitted the manuscript entitled “Synthesis and Chemopreventive Potential of 5-FU/Genistein Hybrids on Colorectal Cancer Cells” to the journal “Pharmaceuticals” in order to be considered for publication as an “Article”.
The presented work reports on the chemical synthesis and analytical characterization of a series covering eight hybrids of 5-FU and genistein with different lengths of their alkyl spacers. For the biological assessment, the authors used two colon carcinoma cell lines and two non-cancerous ones. Cytotoxic and antiproliferative effects were studied, as well as impact on the cell cycle and induction of apoptosis. Docking studies were carried out to support the outcomes from the biological evaluation.
The authors provide a comprehensive manuscript which generally fits the scope of the special issue “Hybrid Drugs: Design and Applications”. However, there a some concerns and suggestions I wish to mention before reconsidering for potentialpublication.
Caption of Figure 1: …derived from 5-FU or genistein…
Caption of Scheme 1: The abbreviations used in the scheme should be explained in the caption. Although they are common to me and probably the most scientists reading the manuscript, providing the key of the abbreviations is more reader-friendly.
Synthesis: What was the reason for omitting the derivatives with n = 5, 8, 9? Please provide an explanation in the manuscript.
Can the authors provide a reasonable explanation in their manuscript why the linker consisted of a 1,2,3-triazol moiety? What is the biological impact of this moiety? Or was it just selected due to the click-chemistry approach?
Line 125-126: Maybe it is better to term it “GI50” value?
Maybe exchange “cytotoxic activity” for “cytotoxicity”.
Table 1: column on the very right – it must be SW-620; 4a/CHO-K1 – 71.10 ± 10.50 and 5-FU/HaCaT – 118.7 ± 2.8 (take care about significant digits)
As far as I get it, the authors investigated the growth inhibition after treatment of the (non-)cancer cell lines with their compounds, and with each Genistein and 5-FU, respectively. However, I am wondering about the effect when adding both “unconnected” compounds Genistein and 5-FU (kind of physical combination). The authors are kindly asked to add these data, since this would help getting evidence, if there is an additive or even a synergistic effect by their hybrids.
Line 147: …through sulforhodamine B assay.
“The antiproliferative activity of the most active hybrids was evaluated against SW480 (4a) and SW620 (4a and 4g) colon cancer cells” I am wondering: why was compound 4f showing better IC50 values against SW620 excluded from further investigation but compound 4a having nearly equal (slightly poorer activity) included? Besides, the SI does not differ significantly. The authors are kindly asked to add experiments or to provide a very reasonable explanation in their manuscript.
“…induced changes in the morphology of the cells, altering size and shape, causing a severe damage, suggesting a process of death.” Was it possible to detect hallmarks of apoptosis, such as chromatin condensation, shrinkage, blebbing, formation of apoptotic bodies?
Table 2: The caption of the table should include information about the representation of the data, i.e. mean ± SD/SE of n=? independent experiments. However, I suggest to remove this table to the supporting information as Figure 3 presents the same data but just in a more appealing way.
“…since it intercalates into the major groove of double-stranded DNA.” However, maybe it should mentioned that PI is used to stain necrotic cells (no membrane integrity), while Annexin V dyes apoptotic cells (intact membranes) in contrast.
Figure 4: Exchange “comma” and use “full stop” instead when reporting on values. Please add to the caption that it is about compound 4a. Both aspects also apply for Figure 5. Please revise.
Line 297: loose?
Line 318: independent of?
Line 328: caspases
Line 502: shown
Figure 10: Fluorine
“…NMR spectra were obtained on a Varian equipment.” Please provide more information about the spectrometer.
Line 590: TMS (explain abbreviation) tetramethylsilane … among other abbreviations
Line 758, etc.: change “comma” and use “full stop” instead when reporting the molecular masses.
How about the purity? At least the authors must confirm purity of > 95% of the end-products 4a-h either using CHN-analysis or HPLC. Please provide these data.
Generally, I missed a Supplementary Information that includes the spectra, e.g. from the 1H and 13C NMR analysis. Please provide these spectra in the Supplementary Material.
@ 3.2.1 how about regularly testing for mycoplasma infection of the cell lines?
Line 855: 20,000
Line 945: μM
Line 946: 5-FU
The authors are kindly asked to consider effects of the hybrids, especially of their most studied derivative 4a on COX-2 as this was considered to be influenced by 5-FU and genistein: 10.1016/j.bbrc.2005.04.143.
Can the authors please provide in their manuscript a comment on the role of the spacer-length for future design. It seems that only the shortest spacer is valuable.
Although there will be a close editing by the publisher in case of acceptation of the manuscript, the authors are kindly asked to correct/improve formal errors/inconsistency, such as: caption (line 32, 76, 89, 95, 743), exchange “over” (54, 58), bold (58, 68, 98, 105, 147, 201, 295/296, 370, 389, 447, 457, 476, 481, 485, 488, 495, 496, 509, 510, 512, 514, 519, 521, 523, 536, 539, 550, 554, 567, 569, 570, 572, 580, 581, 607, 613, …(also all other compounds), 664, 669, 737, 869, 903, 904, 920, 944, 947, 950, 952, 956, 958), space (67, 68, 69, Scheme 1, 140, 151, 282, 600, 665, 728, 730, 738, 740, 846, 857, 884, 893), subscript (53, 58, 608, 610, …(also all other compounds), 857, 860, 862), molecular mass (111), superscript (113, 460/461, 588, 589, 608, 610, …(also all other compounds), 882), punctuation (222, 286, 321, 326), in italics (607, 613, …(also all other compounds)). This listing is not to be understood completed but shall give the authors some information about what to correct/improve.
Author Response

(The authors gave the same response as above.)

Round 2
Reviewer 2 Report
The authors Gustavo Moreno-Q, Wilson Castrillón-L and co-workers provided a revised version of their manuscript. They did a diligent job to consider the comments, however, in my opinion there are still some aspects. Embedding this would make the work of the authors more reasonable for the readers.
# It is suggested to list the abbreviations in an alphabetical order.
# The reason for omitting the derivatives with n = 5, 8, 9 (import problems) was not provided in the manuscript. Or did I miss it? The same applies for the reason of choosing the 1,2,3-triazol moiety as a linker and their decision selection of the compounds for further testing.
# Table 1: Using the physical combination of 5-FU and Genistein, the authors mentioned that the mixture is equimolar but the do not provide a real concentration in the caption of the table. Die physical mixture reaches lower IC50 values compared to all compounds 4a-h in all investigated cell lines (exception 4g, SW620: similar effect) meaning higher cytotoxicity, even higher than each of the two compounds 5-FU and Genistein. The authors must provide a reason why their compounds are nevertheless useful, i.e. derivatives bearing covalent connected 5-FU and Genistein.
# The interesting aspect on the combination of 5-FU and Genistein on COX-2 (even a reference was provided) was not considered in the revised manuscript although this would somehow justify the work of the authors, i.e. somehow combining 5-FU and Genistein.
# HPLC analysis: 4a: why is the peak at retention time about 10 min not considered for integration? 4h: why are the peaks at retention time about 8.5 min and 14 min not considered for integration? The chromatograms confirming purity >95% of the other compounds of the series 4a-h are missing. HPLC analysis of all compounds subject to biological testing must be secured and therefore performed before doing the biological experiments. Therefore, it should be not a big deal for the authors to provide the chromatograms.
Author Response
Dear reviewer,
Thank you in advance for reviewing the manuscript entitled " Synthesis and Chemopreventive Potential of 5-FU/Genistein Hybrids on Colorectal Cancer Cells". In this regard, I inform you that it was corrected according to your valuable observations. The modifications can be found highlighted in the new version of the document to facilitate the review process.
Thank you so much.
Kind regards,
______________________
Wilson Cardona-G
Chemistry Institute
Universidad de Antioquia
Colombia

Round 3
Reviewer 2 Report
Unfortunately, you have again included the same response letter (dated September 16, 2022) as the last revision. Could you please attach the current response letter? You will probably also need to upload the revised version of the manuscript. Please check this again.
Thank you very much and best regards!
Author Response
Dear reviewer,
Thank you in advance for reviewing the manuscript entitled " Synthesis and Chemopreventive Potential of 5-FU/Genistein Hybrids on Colorectal Cancer Cells". In this regard, I inform you that it was corrected according to your valuable observations. The modifications can be found highlighted in the new version of the document to facilitate the review process. In addition, the English was revised.
Thank you so much.

Round 4
Reviewer 2 Report
The authors have provided a revised version of their manuscript. In doing so, they have addressed all of the previously mentioned concerns and suggestions. Everything fits so far with one exception, which is not at all respectable. The authors have included the HPLC chromatograms of all compounds in the supplementary material, but the way of integration is not possible in my opinion. In particular, my criticism refers to the peaks at 6 min for compound 4e and about 8.75 min for compounds 4d, 4f, 4g, respectively. There is a clear and constant baseline in the range of 7-12 min, so in my opinion the integration should be included as an imaginary extension of the straight baseline (parallel to the x-axis), so to speak. The authors should correct this, otherwise it looks strange in my opinion. The new purities are to be calculated. Unfortunately, it seems to me that then, however, a purity of >95% is no longer granted. This is actually an indispensable criterion when compounds are subjected to a biological evaluation. I see insufficient purity as problematic.
Author Response
Dear Reviewer Thank you very much for your valuable comments and corrections.
We still have active compound and what we did was to inject it again in the chromatograph with the same method but increasing this time the concentration of the analyte because as seen in the first chromatogram the detector response was very low less than 20 mAU and the additional peak at the retention time about 8.75 min corresponded to impurities present in the mobile phase as annex in the support chromatogram. This additional signal is only less than 1 mAU. When increasing the analyte concentration this peak disappears due to improved signal to noise ratio. Based on these results and with the comparison of the chromatogram of the mobile phase we can guarantee that our analyte has a purity higher than 95%. Thank you for your time and consideration of this support material.

Round 5
Reviewer 2 Report
Thank you to the authors for their quick reply and explanation of the HPLC chromatograms. I hope the information given is correct not only for the compound 4g shown, but also for the others. The peak of 4b seems to have a shoulder and for 4e you could integrate the peak at retention time 3.1 min even closer to the imaginary baseline. But I suggest to leave that now. Thanks to the authors for the patience and all corrections. All the best!